# Selenium May Be Involved in Esophageal Squamous Cancer Prevention by Affecting GPx3 and FABP1 Expression: A Case-Control Study Based on Bioinformatic Analysis

**DOI:** 10.3390/nu16091322

**Published:** 2024-04-28

**Authors:** Niannian Wang, Da Pan, Xiaopan Zhu, Xingyuan Ren, Xingyi Jin, Xiangjun Chen, Yuanyuan Wang, Ming Su, Guiju Sun, Shaokang Wang

**Affiliations:** 1Key Laboratory of Environmental Medicine and Engineering, Ministry of Education, Department of Nutrition and Food Hygiene, School of Public Health, Southeast University, Nanjing 210009, China; wangnian-nian@foxmail.com (N.W.); pantianqi92@foxmail.com (D.P.); zhuxiaopandhr@163.com (X.Z.); mn13451537048@126.com (X.R.); xingyijin@foxmail.com (X.J.); cxjll910@163.com (X.C.); 230218460@seu.edu.cn (Y.W.); gjsun@seu.edu.cn (G.S.); 2Department of Public Health, School of Medicine, Xizang Minzu University, Xianyang 712000, China; 3Huai’an District Center for Disease Control and Prevention, Huai’an 223001, China; suminglala@163.com

**Keywords:** esophagus cancer, esophageal precancerous lesions, selenium, GPx3, FABP1

## Abstract

The role of selenium in the developmental process of esophageal cancer (EC) requires further investigation. To explore the relationship between selenium-related factors and EC through bioinformatic analysis, a case-control study was conducted to verify the results. Utilizing the GEPIA and TCGA databases, we delineated the differential expression of glutathione peroxidase 3 (GPx3) in EC and normal tissues, identified differentially expressed genes (DEGs), and a performed visualization analysis. Additionally, 100 pairs of dietary and plasma samples from esophageal precancerous lesions (EPLs) of esophageal squamous cancer (ESCC) cases and healthy controls from Huai’an district, Jiangsu, were screened. The levels of dietary selenium, plasma selenium, and related enzymes were analyzed using inductively coupled plasma mass spectrometry (ICP-MS) or ELISA kits. The results showed lower GPx3 expression in tumor tissues compared to normal tissues. Further analysis revealed that DEGs were mainly involved in the fat digestion and absorption pathway, and the core protein fatty acid binding protein 1 (FABP1) was significantly upregulated and negatively correlated with GPx3 expression. Our case-control study found that selenium itself was not associated with EPLs risk. However, both the decreased concentration of GPx3 and the increase in FABP1 were positively correlated with the EPLs risk (*p* for trend = 0.035 and 0.046, respectively). The different expressions of GPx3 and FABP1 reflect the potential of selenium for preventing ESCC at the EPLs stage. GPx3 may affect EC through FABP1, which remains to be further studied.

## 1. Introduction

Esophageal cancer (EC) has garnered significant attention within the global cancer landscape due to its high incidence and mortality rates [1]. The absence of early and mid-stage neoplasm diagnosis, owing to its inconspicuous nature, poses significant challenges to achieving a favorable prognosis or managing mortality rates [2]. In 2022, the World Health Organization (WHO) reported that, globally, approximately 511,000 cases of EC were recorded, placing it as the 11th most common cancer worldwide. Additionally, about 445,000 deaths were attributed to EC, ranking it seventh in terms of mortality rates [3]. China, a country with a high incidence of EC, reported over 187,000 deaths in 2022, making it the 5th leading cause of cancer mortality within the country [4]. Given these statistics, it becomes evident that the primary prevention of EC is crucial. Widespread recognition has been accorded to the reduction of EC risk through dietary and living-environment improvements [5].

Selenium is an essential trace element that can be detected in soil, water, and air [6]. It enters the food chain through bioaccumulation and serves as a primary route for selenium supplementation in humans [7,8]. Many studies have demonstrated the significant effect of selenium on EC prevention. Ahsan et al. [9] found that selenium exhibits anti-inflammatory and antioxidant effects, inhibiting the proliferation of EC cell lines and retarding its development in mice. In addition, long-term selenium supplementation has shown positive effects on gene expression associated with inflammation, cell proliferation, and apoptosis in normal esophageal tissues of rats [10]. Cohort studies also can be found. Several separate studies from Iran found that serum selenium levels were significantly lower in EC patients than in controls. Meanwhile, the concentration of selenium was lower in EC tissues compared to non-cancerous tissues [11,12,13]. A meta-analysis by Cai et al. [14], which comprised five studies, showed that high selenium exposure may reduce the EC risk.

Selenium is normally bound to proteins in vivo and is involved in physiological activities in the form of selenoproteins [15]. A deficiency of selenium leads to low levels of antioxidant selenoproteins [16,17]. Glutathione peroxidase (GPxs) is a family of antioxidant enzymes with eight members, GPx1–GPx8 [18]. The human GPx3 gene is located on chromosome 5 and consists of five exons. GPx3 is an extracellular glycoprotein, and therefore, it can be considered an efficient extracellular device against oxidative stress [19]. As the only extracellular isoform of the family, GPx3, can circulate and be detected in plasma [20]. Some studies have found that selenium levels in vivo have a significant positive correlation with GPx3, leading to its widespread utilization as a biomarker [21,22].

The downregulation of GPx3 expression is closely associated with cancers. Several studies have shown reduced serum GPx3 activity in cancer patients [23,24]. In addition, studies utilizing GPx3 KO mice have shown accelerated dysplasia in tumor lesions and an increased incidence of cancer [25,26]. However, studies focusing on GPx3 and its relationship with EC remain limited. A case-control study showed that the expression level of GPx3 in normal esophageal epithelial tissues was three times higher than that in EC tissues, indicating that the downregulation of GPx3 serves as a significant indicator of EC development [27]. However, the bioinformatic analysis involved in this study did not delve into the possible causes of GPx3 downregulation to support the experimental results. Another bioinformatic analysis also clarified that GPx3 expression was downregulated in EC tissues. An attempt was made to find differentially expressed genes (DEGs), but no experimental data were available to further validate [28].

We hope to clarify the role of selenium and related metabolites in reducing EC incidence through bioinformatic analyses to support health promotion. However, EC patients have lost the opportunity to undergo primary and secondary prevention. Therefore, studies targeting esophageal precancerous lesions (EPLs) should be highlighted. EPLs often progress to EC in situ following deterioration, but appropriate intervention can slow down or even reverse this process [29,30]. Meanwhile, a study found that genetic mutations and copy-number alterations observed in EPL tissues closely resemble those found in EC tissues [31]. Therefore, it is feasible to validate the expression of dietary selenium intake and related proteins in patients with EPLs based on bioinformatics analyses, which, at the same time, has greater relevance for the prevention of EC.

EC comprises two primary subtypes: esophageal squamous cell carcinoma (ESCC) and esophageal adenocarcinoma (EAC). ESCC is prevalent in Asia, South Africa, and South America, with major contributing factors including smoking, alcohol consumption, food choices, and eating patterns [32]. EAC is the main histological type of EC in the West, including North America, Europe, and Australia. Huai’an district in Jiangsu province is one of the high-risk areas of ESCC in China. The total number of new cases of ESCC in this region from 1998 to 2016 was 20,892, accounting for 38.01% of the new cases of cancer during the same period, both of which ranked first in the ranking of malignant tumors in the region [33].

Our study explored the possible association of GPx3 with EC and its underlying mechanisms systematically through DEG screening, GO and KEGG pathway enrichment analysis, and PPI visualization. Subsequently, a 1:1 case-control study about EPLs in ESCC from Huai’an district, Jiangsu, was used to validate the results of the bioinformatic analysis and to explore the effects of selenium and related proteins on the risk of EPLs and to explore the effects of selenium and related proteins on the risk of EPLs.

## 2. Materials and Methods

### 2.1. Bioinformation Analysis

The Gene Expression Profiling Interactive Analysis (GEPIA) database (http://gepia.cancer-pku.cn/, accessed on 28 April 2022) was used to analyze the differential expression of GPx3 between tumor tissues and normal tissues. The EC dataset was selected, and we set the |Log_2_FC| cutoff to 2 and the q-value cutoff to 0.01. GPx3 was retrieved and more detailed gene expression was obtained.

The RNA sequence data and clinical information on esophagus cancer were downloaded from The Cancer Genome Atlas (TCGA) database (https://www.cancer.gov/ccg/research/genome-sequencing/tcga, accessed on 30 April 2022). The exclusion criteria were as follows: (a) without clinical or prognostic information, (b) other malignancies in addition to EC, and (c) patients without follow-up data or overall survival < 30 days.

The sample data were analyzed using the “edgeR”, “limma” and “gplot” packages in R. The fold change (FC) and the corresponding *p* value were considered indicators for DEGs. We set log|FC| > 2.0 and *p* < 0.01 as significant. The DAVID database can provide biofunctional annotation and analysis for large-scale genes. All selected DEGs were ID-transformed by DAVID, and GO and KEGG enrichment analysis and mapping were performed based on the “clusterProfiler” and “ggplot2” packages in R. STRING was used to pool selected DEGs for protein–protein interaction (PPI) network identification. The results were exported as images to Cytoscape v3.8.2 for visual analysis, and the top-ten hub genes were clustered using the CytoHubba plugin for display.

GEPIA was used to analyze the correlation of GPx3 and FABP1. *p* < 0.05 (two tailed) were considered statistically significant.

### 2.2. The Case-Control Study

#### 2.2.1. Population-Sample Collection

The recruitment of subjects came from The Early Diagnosis and Early Treatment Project of Esophageal Cancer (EDETPEC) [34,35]. The study was conducted according to the guidelines of the Declaration of Helsinki and approved by the Institutional Review Board of Southeast University Zhongda Hospital (Nanjing, China) with an approval number of 2012ZDllKY19.0. Beginning in 2011, EDETREC recruited subjects through face-to-face questionnaire interviews by village healthcare workers. The subjects were in the age range of 35 to 75 years and did not have a history of any kind of cancer. Participants underwent a routine endoscopy examination, tissues were biopsied, and diagnosed according to the histologic criteria of dysplasia. Based on this, 1731 residents who had lived in the towns of Jibiao, Jingkou, and Qiuqiao for more than 5 years were recruited between January 2015 and June 2017 for this study. The routine endoscopy examination revealed that 148 of them were diagnosed with EPLs.

The recognized EPL of ESCC is esophageal squamous dysplasia, which usually occurs in the mucosa of the esophagus. The WHO considers EPLs in ESCC to be esophageal squamous cells characterized by cytological atypia (nuclear atypia: enlargement, pleomorphism, hyperpigmentation, loss of polarity, and overlapping) and structural atypia (abnormal epithelial maturation). The WHO classifies EPLs as low and high grade. Involvement of only the lower half of the epithelium, with only mild cytological atypia was classified as a low-grade EPL, whereas high grade meant involvement of more than half of the epithelium or severe cytological atypia (regardless of the extent of epithelial involvement) [36].

A total of 144 residents were diagnosed with low-grade EPLs, of which 100 cases were randomly selected. One hundred cases were selected as healthy controls from 1583 healthy residents, matched by sex, age (±2 years), and village. The epidemiological data and dietary information obtained from the survey have been published [34]. We have also previously described in detail the collection and preservation of the dietary and plasma samples from the 200 subjects [35].

#### 2.2.2. Laboratory Measurements

The dietary and plasma samples need to be filtered and purified before being tested for indicators (Appendix A). Inductively coupled plasma-mass spectrometry (ICP-MS) (Agilent Technologies (China) Co., Beijing, China) was used to determine the levels of selenium in the dietary and plasma samples. Under the optimal operating conditions of the instrument, the blank control sample, the mixed standard working solutions, and the diluted sample solutions for measurement were introduced into the ICP-MS by the injection system and peristaltic pump. The selenium contents were calculated from the measured abundance values of the samples corresponding to the standard curve automatically plotted by the instrument.

The expression concentrations of GPx3 and FABP1 in plasma samples were determined by ELISA kits (Nanjing Jiancheng Bioengineering Institute, Nanjing, China). The assay was performed in strict accordance with the manufacturer’s recommended procedure. The OD values were measured at 450 nm using an enzyme marker (Tecan Trading Co., Ltd., Shanghai, China).

#### 2.2.3. Statistical Analysis

A Wilcoxon rank-sum test was conducted to analyze the results of indicator measurements. A Spearman correlation analysis was used to understand the correlation of plasma selenium levels with dietary selenium content, GPx3, and FABP1 in patients with EPLs and healthy subjects. Conditional logistic regression was performed to evaluate the association between selenium-related variables and the risk of EPLs, with adjustment for age, gender, body mass index (BMI), education level, annual income, tobacco smoking, alcohol drinking, tea drinking, meals on time, spicy taste acceptance, family history of esophagus cancer, and history of digestive diseases. Selenium-related variables were triangulated to complete subsequent analyses. All covariates were categorical except age and BMI, which were continuous variables. For better analysis, BMI was set as a dummy variable. It was classified based on the criteria from the WHO (http://apps.who.int/gho/data/node.main.BMIANTHROPOMETRY?lang=en, accessed on 30 April 2022), with underweight (BMI < 18.5), normal (BMI 18.5–24.9), overweight (BMI 25–29.9) and obese (BMI ≥ 30). Each variable was analyzed by way of “enter”. *p* < 0.05 (two-tailed) were considered statistically significant. All statistical analyses were performed on SPSS (version 22.0). The statistical review of the study was performed by a biomedical statistician.

## 3. Results

### 3.1. Expression of GPx3 in Tumor and Normal Tissues

A total of 182 tumor tissues of EC and 286 normal tissues were retrieved from GEPIA to analyze the differential expression of GPx3. The median gene expression of GPx3 in cancer tissues was 38.489, which was significantly lower compared to 401.971 in normal tissues. As shown in Figure 1, the expression of GPx3 was significantly lower in the tumors than in the controls.

### 3.2. Visualization of DEGs in the Tumor Tissues with Low Expressed GPx3

A total of 162 tumor tissues were searched from the TCGA. The samples were divided into two groups using the median of the GPx3 expression levels as a limit. As shown in Table 1, there were no significant differences in age and EC stage between the two sample groups of GPx3.

A total of 165 DEGs were demonstrated, of which 121 were downregulated genes and 44 were upregulated ones. (Appendix A and Figure 2A). GO and KEGG were responsible for the enrichment analysis of the DEGs. As shown in Figure 2B, most of the DEGs are located at the apical part of the cell in tissues with low expression of GPx3. In addition, their main function is to participate in receptor ligand activity and the process of epidermal cell differentiation. The DEGs are mainly involved in fat digestion and the absorption pathway.

The top-10 hub genes ranked were screened based on String online analysis and visualized by Cytoscape. The most central protein, fatty acid-binding protein 1 (FABP1), was found to be upregulated in cancer tissues with low GPx3 expression (Figure 2C).

We further analyzed the correlation between GPx3 and FABP1 using GEPIA to validate the DEG visualization results. Figure 3 shows that there was a significant negative correlation between GPx3 and FABP1 (*p* = 0.012).

### 3.3. Selenium-Related Variables of the Subjects

As shown in Table 2, no statistical difference in the daily dietary intake of selenium was found between the two groups. However, the plasma selenium level of EPL cases was significantly lower than that of healthy controls (*p* < 0.001). The protein concentrations of GPx3 did not differ between the two groups. A further detection, however, revealed that the EPLs group had higher concentrations of FABP1.

We found that dietary selenium intake was significantly positively correlated with plasma selenium, suggesting that exogenous supplementation of selenium can change selenium levels in vivo. In addition, there was also a positive correlation between plasma selenium levels and GPx3 concentrations, which adds to the evidence that GPx3 can serve as a biomarker of selenium status in vivo. Although FABP1 expression was negatively correlated with plasma selenium in both groups, neither was significant (Table 3).

### 3.4. Correlation between Selenium-Related Variables and Risk of EPLs

As shown in Table 4, our study did not find an association between a daily dietary intake of selenium and EPLs risk (*p* for trend = 0.545). The concentration of plasma selenium located in the highest tertile was associated with a reduced risk of EPLs compared to that in the lowest tertile (OR (95% CI) = 0.34 (0.13–0.93)). In addition, the concentration of GPx3 was negatively associated with the EPLs risk (*p* for trend = 0.035).

However, the risk of EPLs increases with higher protein concentrations of FABP1 (*p* for trend = 0.04). This is consistent with the results of our previous bioinformatic analyses.

## 4. Discussion

GPx3 expression was lower in tumor tissues compared to normal tissues. DEGs were found to act mainly in the fat digestion and absorption pathway, and the upregulation of the most central hub protein, FABP1, was also associated with low GPx3 expression. Subsequent case-control studies supported these results. We further found that although selenium was not associated with EPLs, the reduced concentration of GPx3 and the upregulation of FABP1 increased the risk of EPLs.

The reduced GPx3 expression in the plasma of EPL cases may be related to epigenetic regulation. Nirgude et al. [37] highlighted GPx3 mutations and copy-number alterations in cancers through TCGA analysis. In addition, there was hypermethylation of the CpG island within the GPx3 promoter in ESCC cell lines. Moreover, the GPx3 promoter exhibited more frequent methylation in ESCC tissues compared to adjacent normal tissues [38,39]. Consequently, this methylation pattern leads to the downregulation of both mRNA and protein expression of GPx3 in EC [27]. Secondly, GPx3 was shown to be effective in slowing down the proliferation, migration, and invasion of lung cancer cells under oxidative stress and in significantly reducing the production of ROS [40,41]. Meanwhile, GPx3 silencing resulted in ROS production, DNA damage, and increased apoptosis in Caco2 cells [25]. This suggests that GPx3 deficiency is associated with enhanced lipid peroxidation [42,43].

Lipid peroxides serve as pivotal agents in cancer by disrupting the body’s homeostasis through REDOX balance disturbances [44]. As we analyzed with KEGG, FABP1 played an important role in the fat digestion and absorption pathway. This is due to the presence of fatty acid-binding cavities within the FABP1 protein that facilitate fatty acid uptake and transport [45]. Wu et al. [46] found that the overexpression of FABP1 significantly increased fatty acid uptake in hepatocytes. Therefore, FABP1 is considered a protective antioxidant molecule that controls the accumulation of peroxidation by inducing the inactivation of active lipids [47]. The thiobarbituric acid-reactive substances (TBARS) are the marker of lipid peroxidation [48]. FABP1 KO mice were found to have significantly reduced levels of TBARS in the liver, along with a decreased expression of markers of inflammation and oxidative stress [49]. The absence of GPx3 in EPLs may lead to the increase of TBARS, thus stimulating the increase of FABP1 expression to coordinate the oxidative stress state of the body. Therefore, our database analysis revealed a significant upregulation of FABP1 in EC tissues with low GPx3 expression. Meanwhile, our case control also found a significant increase in FABP1 expression in the plasma of EPLs with downregulated GPx3 expression.

In addition to the manifestations of epigenetic regulation of GPx3 in ESCC, it is possible that selenium status in vivo may also influence its expression. However, neither dietary selenium nor plasma selenium levels were associated with the risk of EPLs, despite demonstrating a positive correlation between selenium and GPx3. A meta-analysis conducted by Vinceti et al. [50] concluded that the incidence and mortality of several cancers, including EC, were not associated with selenium supplementation or its in vivo status. This relatively strong evidence supports our results.

The possible contribution of selenium to reducing the incidence of ESCC was mentioned previously, and our non-predicted results may be related to the differences in environmental selenium concentrations. The concentration of selenium in soils varies with the type, texture, and organic matter content of the soil, as well as with local rainfall. Like plants, it is influenced by the physicochemical properties of the soil (redox state, pH, and microbial activity). The average concentration of selenium in soils ranges from 0.1 to 0.7 mg/kg. It ranges from 0.8 to 2 mg/kg in clay soils and up to 2 to 4.5 mg/kg in tropical soils. In addition, the presence of selenium in the atmosphere is related to natural and anthropogenic activities. Soil erosion, volcanic activity, and waste incineration all contribute to changes in atmospheric selenium concentrations [6]. It is not difficult to reason that differences in selenium levels in soil and atmospheric deposits can lead to different selenium concentrations in water. The selenium present in diverse soils and water sources can be partially absorbed by crops and ingested by bred animals, subsequently entering the human body. Consequently, the selenium content of plants used as food or feed in different environments may vary significantly. Health problems associated with selenium may be altered due to differences in the body’s tolerance to selenium. A U-shaped nonlinear dose–response relationship has been demonstrated, indicating that both selenium-deficient and selenium-enriched states can be detrimental to health [51]. Therefore, disparities in selenium status due to natural exposure in different habitations might influence overall health conditions [52,53,54,55].

In addition to natural levels of selenium exposure, Hurst et al. [39] found that cancer risk is related to the body’s ability to absorb and metabolize selenium. Interactions between vitamins and minerals may mask the association between selenium and ESCC [40,41]. For instance, trivalent iron forms complexes with selenium, leading to precipitation and preventing its absorption by intestinal cells. When sulfur concentrations exceed 2.4 g/kg DM, selenium absorption is reduced due to competition [56,57].

Furthermore, the specificity of subjects may lead to differences in EC subtypes, which makes it challenging to study the relationship between selenium and ESCC. While China is a region with a high prevalence of ESCC, the current evidence linking selenium and EC is mostly from European and American populations [58]. In addition, the evidence suggests that, despite consistent nationality, ethnic heterogeneity still produces large differences in both the incidence and subtypes of EC [59]. This diversity may be related to variations in dietary patterns, lifestyle habits, and the genetic genes related to selenium absorption and metabolism across different ethnic groups [5,60,61].

Our study also has limitations. First, the fluctuation of GPx3 and FABP1 levels during the development of EPLs into EC was not investigated. Barrett’s esophagus (BE) is a kind of EPL that may progress into EAC [62,63]. Evidence highlights that FABP1 was significantly highly expressed in BE, while the expression gradually decreased as the disease worsened until the occurrence of EAC [64,65]. This is despite the fact that a study proved rational drug therapy resulted in the downregulation of FABP1 expression in BE patients [66]. We detected a significant increase in the FABP1 expression level in EPL cases. However, we remain uncertain whether FABP1 expression also gradually decreases until the diagnosis of EC is established. Second, EC is mostly classified as ESCC in China, leading to a lack of sufficient data pertaining to EAC for the validation of bioinformatics results. We acknowledge the necessity for more extensive international collaboration to overcome these limitations. In addition, a deeper exploration is necessary. Although we have found an association between GPx3 and FABP1 based on the fat digestion and absorption pathway, further experiments and analyses, such as investigating their in situ expression, are still needed to clarify the underlying mechanistic reasons for this association.

## 5. Conclusions

Our study did not establish a direct association between selenium itself and ESCC. However, the significant difference observed between GPx3 and FABP1 in the two groups suggested the potential significance of selenium in EC prevention. In addition, FABP1 may serve as a novel biomarker for EC and EPLs in ESCC, but this still needs to be explored by more in-depth studies.

## Figures and Tables

**Figure 1 nutrients-16-01322-f001:**
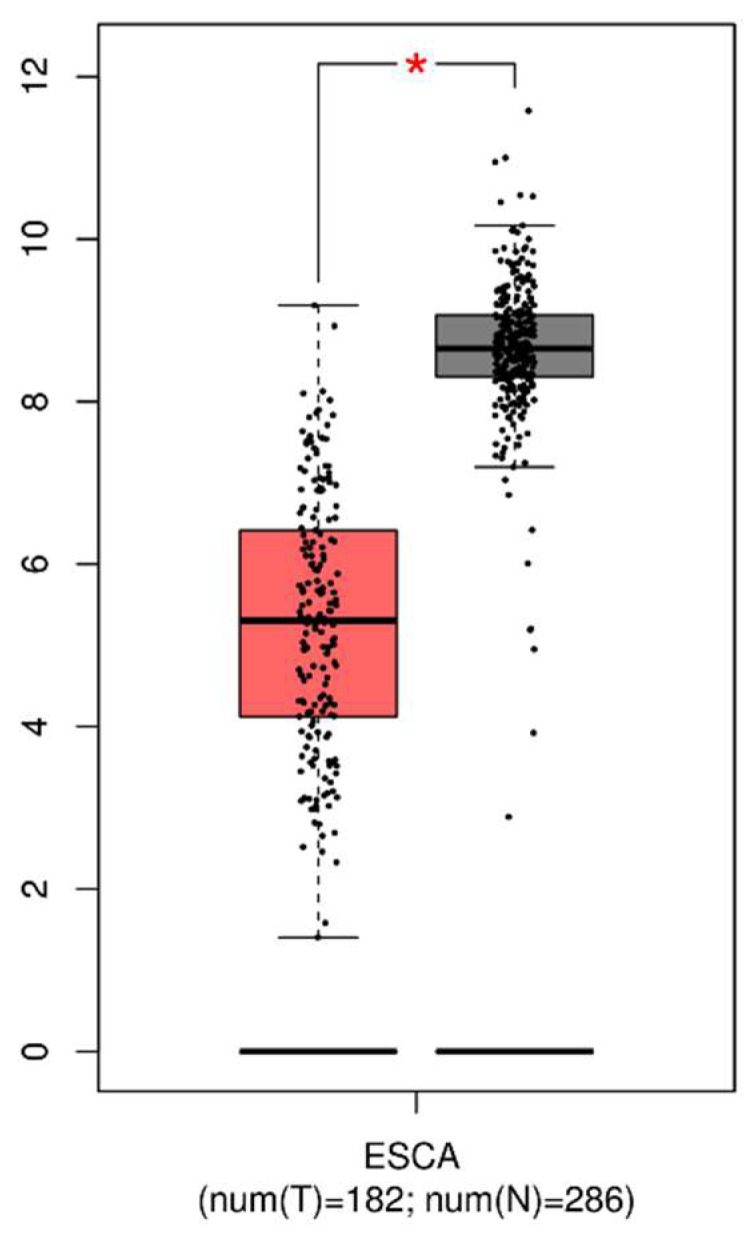
Expression of GPx3 in tumor and normal tissues. For better visualization, GPx3 expressions are log-transformed. We use log_2_(TPM + 1) for log scale. TPM, transcripts per million. * The difference is statistically significant (*p* < 0.05).

**Figure 2 nutrients-16-01322-f002:**
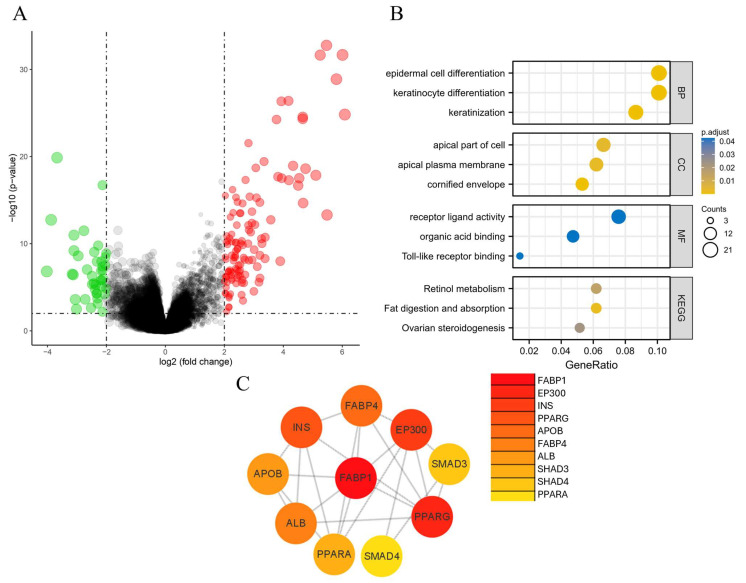
The visualization of the DEGs in low-expression GPx3 vs. high-expression GPx3. (**A**) The volcano plot of the DEGs, in which red dots represent downregulated genes, green dots represent upregulated genes, and black dots show no differences in gene expression; (**B**) GO and KEGG enrichment analysis. (**C**) The top-10 hub DEGs.

**Figure 3 nutrients-16-01322-f003:**
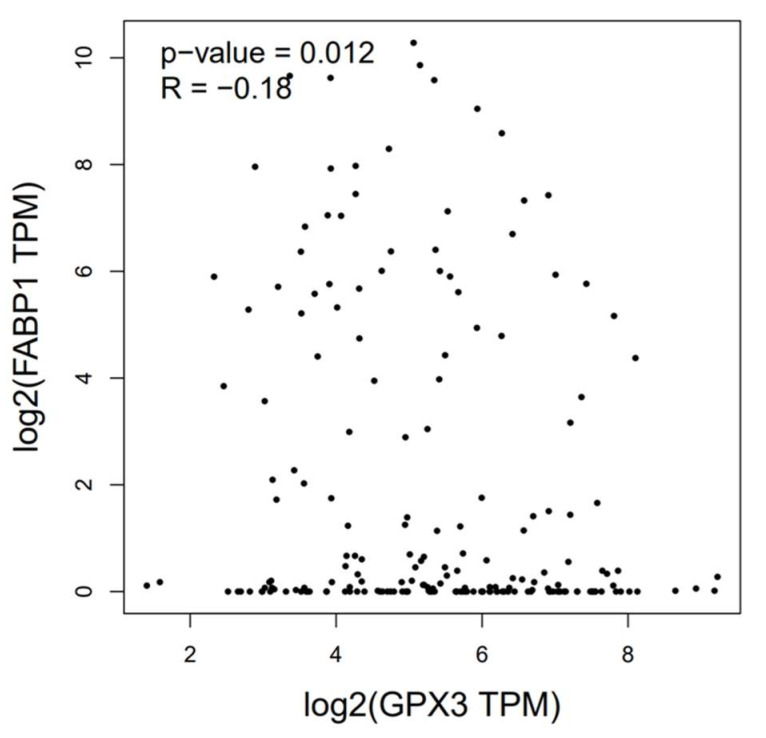
The correlation of GPx3 and FABP1.

**Table 1 nutrients-16-01322-t001:** Characteristics of patients in high and low expression of GPx3.

	Stage	GPx3		
		Low	High	*p* Value
n		81	81	
Age, $\bar{X}$ ± s		62.49 ± 11.79	61.99 ± 12.31	0.790
T, n (%)	T1	11 (7.6)	16 (11)	0.478
	T2	19 (13.1)	18 (12.4)	
	T3	42 (29)	35 (24.1)	
	T4	1 (0.7)	3 (2.1)	
N, n (%)	N0	35 (24.3)	31 (21.5)	0.272
	N1	30 (20.8)	33 (22.9)	
	N2	3 (2.1)	6 (4.2)	
	N3	5 (3.5)	1 (0.7)	
M, n (%)	M0	58 (45)	63 (48.8)	0.486
	M1	5 (3.9)	3 (2.3)	

**Table 2 nutrients-16-01322-t002:** Comparison of selenium-related variables between two groups.

Median (25th–75th)	EPLs	Controls	*p* Value
Dietary samples (n)	100	100	
Dietary Selenium (μg/d)	136.72 (94.77–180.06)	147.20 (99.20–177.65)	0.498
Blood samples (n)	100	100	
Plasma Selenium (μg/L)	49.46 (41.75–57.71)	55.97 (43.07–68.23)	<0.001 **
GPx3 (pmol/mL)	42.78 (35.68–51.18)	44.82 (34.48–56.56)	0.561
FABP1 (ng/L)	1433.61 (1337.08–1512.72)	1388.40 (1297.61–1481.82)	0.011 *

* The difference is statistically significant (*p* < 0.05). ** The difference is statistically significant (*p* < 0.01).

**Table 3 nutrients-16-01322-t003:** Correlation between plasma selenium and related variables between the two groups.

	Plasma Selenium	Dietary Selenium	GPx3	FABP1
EPLs	1.00	0.931 **	0.141 *	−0.76
Controls	1.00	0.924 **	0.239 *	−0.82

* The difference is statistically significant (*p* < 0.05). ** The difference is statistically significant (*p* < 0.01).

**Table 4 nutrients-16-01322-t004:** ORs (and 95% CIs) for selenium-related variables with EPLs.

	Q1	Q2	Q3	*p* for Trend
	OR (95% CI)	OR (95% CI)	OR (95% CI)	
Dietary samples				
Dietary Selenium (μg/d)	11.44–114.27	114.28–165.68	165.69–379.39	
EPLs cases (%)	48.48	52.24	49.25	
crude model	1.00	1.22 (0.56–2.67)	1.05 (0.49–2.24)	0.846
adjusted model #	1.00	1.91 (0.69–5.29)	1.45 (0.52–4.01)	0.545
Blood samples				
Plasma Selenium (μg/L)	5.03–45.53	45.54–59.26	59.27–115.92	
EPLs cases (%)	56.06	58.82	34.85	
crude model	1.00	1.29 (0.63–2.63)	0.39 (0.19–0.82) *	0.021 *
adjusted model	1.00	2.04 (0.79–5.25)	0.34 (0.13–0.93) *	0.103
GPx3 (pmol/mL)	17.00–37.79	37.80–47.99	48.00–125.96	
EPLs cases (%)	52.24	61.19	36.36	
crude model	1.00	1.64 (0.78–3.47)	0.56 (0.27–1.13)	0.291
adjusted model	1.00	1.56 (0.60–4.04)	0.27 (0.10–0.70) *	0.035 *
FABP1 (ng/L)	1192.84–1346.47	1346.48–1465.23	1465.24–1649.36	
EPLs cases (%)	73.68	52.24	55.22	
crude model	1.00	1.41 (0.73–2.71)	1.54 (0.82–2.90)	0.180
adjusted model	1.00	1.77 (0.77–4.09)	2.30 (0.99–5.31)	0.046 *

* The difference is statistically significant (*p* < 0.05). # The model adjusted for age, gender, BMI, education level, annual income, tobacco smoking, alcohol drinking, tea drinking, meals on time, spicy taste acceptance, family history of esophagus cancer, and history of digestive diseases.

## Data Availability

All data are available with the consent of the corresponding author.

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
