# Peer review of "Selenium May Be Involved in Esophageal Squamous Cancer Prevention by Affecting GPx3 and FABP1 Expression: A Case-Control Study Based on Bioinformatic Analysis"

_nutrients, 2024, doi:10.3390/nu16091322_

Round 1

Reviewer 1 Report

Comments and Suggestions for Authors

Major

Esophageal cancers include Barrett's esophageal adenocarcinoma, which is common in the West, and squamous cell carcinoma, which is common in the East. Therefore the category of esophageal precancerous lesions should be specifically described.

Minor

There are incorrect line breaks.

Author Response

Thank you very much for your comments. We have responded thoughtfully to all of the comments and revised the manuscript accordingly.

Here are our answers to each comment:

Major

Esophageal cancers include Barrett's esophageal adenocarcinoma, which is common in the West, and squamous cell carcinoma, which is common in the East. Therefore the category of esophageal precancerous lesions should be specifically described.

A:

All of the patients with EPLs we included were potential candidates for esophageal squamous carcinoma(ESCC). We have made additions or modifications where appropriate throughout the text.

Minor

There are incorrect line breaks.

A: Line 67 and 296

Already modified.

Reviewer 2 Report

Comments and Suggestions for Authors

Specific comments to the authors

The authors Nian-Nian Wang et al. of the submitted manuscript "Selenium and Esophageal Cancer: A Case-control Study Based on Bioinformation Analysis" investigated the possible association between selenium and oesophageal cancer (EC). To do this, the authors used various in-silico analyses in combination with molecular techniques such as plasma mass spectrometry (ICP-MS) or ELISA.

In conclusion, based on their primary in silico investigations, the authors demonstrated (i) a lower expression of GPx3 in tumour tissues compared to normal tissues, which (ii) was paralleled by DEGs in fat digestion and absorption pathways as well as by an upregulation of the core protein fatty acid binding protein 1 (FABP1). The consecutive case-control study shows no association of selenium per se with esophageal precancerous lesions (EPLs), although both the decrease in GPx3 concentration and the increase in FABP1 were positively correlated with the risk of EPLs.

Overall, the manuscript provides some interesting information and links between selenium and EC. The manuscript (including the presentation) is clear and convincing. The methods are mostly well described. Although the results and discussion are clearly presented, the authors (see specific comments) need to make some major changes (additional analyses, explanations and experiments) to improve the manuscript. In conclusion, the data presented are interesting. After incorporation of the mentioned specific comments (see below), the manuscript has the potential to be accepted.

Major concerns:

The author investigates the selenium biomarker GPx3 in silico in cancer tissue and subsequently the selenium concentration as well as the enzyme activity in the blood of patients with oesophageal precancerous lesions. Therefore, a non-comparable patient population was used for their investigations. Furthermore, the in situ expression of GPx3 and FABP1 in these oesophageal precancerous lesions was not investigated, which limits the impact of the study.

Specific comments

Title: In relation to the title and throughout the manuscript, it is not clear what type of oesophageal cancer is being studied (squamous cell or adenocarcinoma), which is essential to all investigations and understanding. Furthermore, the title does not reflect the nature of the investigations and findings.

Abstract: The statement "Selenium correlates have the potential to prevent and control EC at the stage of EPLs" is not supported by the experimental design and results. A definitive conclusion of the results is missing. Please add appropriate information.

Introduction: Please explain the characteristics or intention of the "GPX3 biomarker" in the sentence "Therefore, the high sensitivity to selenium levels in vivo has led to the widespread use of plasma GPx3 as a biomarker".

Materials and methods: Please explain the term "mild to moderate EPLs" used in relation to the WHO classification of oesophageal precancerous lesions. In addition, please provide information on the input method used for conditional logistic regression.

Results:

# Table 1: Please add information on the cut-off value for low and high GPx3 expression.

# Figure 2: It is very surprising that GEPx3 is not included in the 10 hub DEGs of Figure 2C. Please explain.

# Table 2: Please perform a multivariate analysis to evaluate the "biomarker" power of selenium, GPx3 and FABP1.

Discussion: Please discuss the results and do not repeat them. What is/are the real outcome(s) of the study?

Comments on the Quality of English Language

Minor editing of English language required.

Round 2

Reviewer 1 Report

Comments and Suggestions for Authors

The manuscript was well revised and 

though to be accepted the journal "Nutrients".

Reviewer 2 Report

Comments and Suggestions for Authors

Specific comments to the authors

In the revised version of the manuscript, the authors were able to address the previously mentioned concerns in an overall adequate and convincing manner. Specifically, the authors should amend the legend of Figure 3 to read "The correlation analysis of the in silico expression data for GPx3 and FABP1 revealed a significant negative association". Therefore, the revised manuscript "Selenium may be involved in esophageal squamous cancer prevention by affecting GPx3 and FABP1 expression: a case-control study based on bioinformatic analysis" should be accepted after incorporating the last minor revision.

Comments on the Quality of English Language

Minor editing of English language required.